# Repurposing Penfluridol in Combination with Temozolomide for the Treatment of Glioblastoma

**DOI:** 10.3390/cancers11091310

**Published:** 2019-09-05

**Authors:** Hyungsin Kim, Kyuha Chong, Byung-Kyu Ryu, Kyung-Jae Park, Mi OK Yu, Jihye Lee, Seok Chung, Seongkyun Choi, Myung-Jin Park, Yong-Gu Chung, Shin-Hyuk Kang

**Affiliations:** 1Department of Neurosurgery, Korea University Anam Hospital, Korea University Medicine, Seoul 02842, Korea; 2Department of Neurosurgery, Korea University Guro Hospital, Korea University Medicine, Seoul 08308, Korea; 3Department of Neurosurgery, VHS Medical Center, Seoul 05368, Korea; 4School of Mechanical Engineering, Korea University, Seoul 02841, Korea; 5Divisions of Radiation Cancer Research, Research Center for Radio-Senescence, Korea Institute of Radiological and Medical Sciences, Seoul 01812, Korea

**Keywords:** penfluridol, sphere forming cell, tumor growth, temozolomide, glioblastoma

## Abstract

Despite the presence of aggressive treatment strategies, glioblastoma remains intractable, warranting a novel therapeutic modality. An oral antipsychotic agent, penflurido (PFD), used for schizophrenia treatment, has shown an antitumor effect on various types of cancer cells. As glioma sphere-forming cells (GSCs) are known to mediate drug resistance in glioblastoma, and considering that antipsychotics can easily penetrate the blood-brain barrier, we investigated the antitumor effect of PFD on patient-derived GSCs. Using five GSCs, we found that PFD exerts an antiproliferative effect in a time- and dose-dependent manner. At IC50, spheroid size and second-generation spheroid formation were significantly suppressed. Stemness factors, SOX2 and OCT4, were decreased. PFD treatment reduced cancer cell migration and invasion by reducing the Integrin α6 and uPAR levels and suppression of the expression of epithelial-to-mesenchymal transition (EMT) factors, vimentin and Zeb1. GLI1 was found to be involved in PFD-induced EMT inhibition. Furthermore, combinatorial treatment of PFD with temozolomide (TMZ) significantly suppressed tumor growth and prolonged survival in vivo. Immunostaining revealed decreased expression of GLI1, SOX2, and vimentin in the PFD treatment group but not in the TMZ-only treatment group. Therefore, PFD can be effectively repurposed for the treatment of glioblastoma by combining it with TMZ.

## 1. Introduction

Glioblastoma is the most common primary malignant brain tumor and has highly aggressive features [1]. The standard treatment of glioblastoma consists of a maximal safe resection and concurrent chemo-radiation therapy with the alkylating agent, temozolomide (TMZ) [2]. However, the conventional treatments for glioblastoma have limitations due to the stem-like characteristics and invasiveness of glioma cells, development of therapeutic resistance in glioma cells, and intratumoral heterogeneity [3,4,5]. Therefore, even in the background of methylated O6-methylguanine-DNA methyl-transferase (MGMT) promoter—a favorable prognostic factor in glioblastoma—the 5-year survival rate of glioblastoma is less than 5% [6]. Recently, there have been several clinical trials to overcome the limitations in glioblastoma treatment, but the results were not favorable [7,8].

The resistance of glioblastoma to current therapies is partially but primarily related to glioma sphere-forming cells (GSCs), which proliferate by self-renewal, and repopulate the entire tumor bulk [9,10]. Genomic studies of recurring glioblastoma samples revealed a pattern with respect to the expression of stemness-related genes and associated mutations in glioblastoma recurrence, suggesting that there is self-renewal as well as repopulation of the cells from ancestral cells [4,11]. Therefore, regulation of GSCs is expected to play an important role in the resistance to treatment and recurrence of glioblastoma, and a novel strategy for dealing with GSCs is urgently needed.

It is known that patients with psychiatric diseases, including schizophrenia, have a lower risk of cancer [12,13,14]. Considering the fact that some antipsychotic agents have been shown to have anti-neoplastic effects and that the antipsychotics easily penetrate the blood-brain barrier [15,16], it can be hypothesized that the antipsychotics can be an ideal alternative for use as a repurposing drug to treat glioblastoma. Penfluridol (PFD) is an oral antipsychotic agent that has been used to treat schizophrenia [17]. Recent studies demonstrated that PFD has antitumor effects on various cancers [18,19,20]. However, whether PFD has an effect against GSCs has not been previously investigated.

In this study, we found that PFD effectively reduced the stemness, invasiveness, and epithelial-to-mesenchymal transition (EMT) potential of GSCs. In addition, PFD treatment suppressed GSC-associated factors, including GLI1, SOX2, uPAR, and vimentin. In combination with TMZ, PFD treatment significantly suppressed tumor growth and prolonged survival in an in vivo orthotopic xenograft animal model. Taken together, our data suggest the novel possibility of repurposing PFD for treating glioblastoma.

## 2. Results

### 2.1. The Effect of PFD on GSCs Proliferation

It is well-known that the response of TMZ, which is a standard chemotherapeutic drug in glioblastoma, depends on the expression of MGMT protein. To determine the effect of TMZ on GSCs, we first determined MGMT protein expression in T98G and U87MG glioma cells and six patient-derived GSCs. Western blot revealed that MGMT protein was expressed only in T98G glioma cells (Figure 1A). Next, we examined glioma cells and GSC proliferation following TMZ treatment in vitro (Figure 1B). In the case of T98G and U87MG, the degree of cell proliferation after TMZ treatment was dependent on MGMT expression. The IC50 of U87MG cell was 100 µM at 5 days TMZ treatment, but the T98G cell did not show IC50 dose until 600 µM TMZ treatment. However, GSCs, were resistant to TMZ, and mostly did not reach IC50, even though MGMT protein expression was not observed.

To investigate the growth suppression effects of PFD, we examined the cytotoxicity of PFD in GSCs by MTT assay. Treatment of X01, 0315, CSC2, 83NS, and 528NS cells with PFD resulted in reduced proliferation of cells in a dose- and time-dependent manner (Figure 1C). The IC50 of PFD after 24 h treatment ranged from 3 to 8 μM in all the GSCs tested. The IC50 was reduced to about 1.6 to 4.5 µM after 48 and 72 h treatment, respectively. In combination with TMZ, PFD significantly suppressed the X01, 0315, CSC2 cell proliferation compared to TMZ only regimen (Appendix A).

### 2.2. PFD Suppresses the Stemness of GSCs

To determine the stemness potential of GSCs in the background of PFD treatment, we examined the tumor sphere formation by phase-contrast microscope. After 48 h of PFD (3 µM) treatment, we observed that sphere formation was decreased by 62, 60, and 64% in case of 0315, CSC2, and X01 GSCs, respectively (Figure 2A, *p* ≤ 0.001). By performing limiting dilution assay, we confirmed that PFD treatment inhibited the ability to form spheres in CSC2 cell (Figure 2B). In addition, the number of secondary spheres also reduced with the increasing concentration of PFD in CSC2 and 528NS cells (Figure 2C). To further confirm these observations, we investigated the markers of stemness, including SOX2 and OCT4. RT-PCR analysis revealed that 2 µM PFD treatment reduced the mRNA levels of SOX2, NESTIN, and OCT4 at 48 h (Figure 2D). The western blot also confirmed the suppression of stemness markers, including SOX2, NESTIN, and OCT4 in CSC2 and 0315 GSCs after PFD treatment (Figure 2E). Moreover, the immunofluorescence assay revealed the inhibition of SOX2 by PFD treatment in CSC2 cells (Figure 2F).

### 2.3. PFD Inhibits the Migration and Invasion of GSCs

It is known that one of the typical characteristics of GSCs is tumor cell invasion into the adjacent normal brain parenchyme, which is recapitulated in the animal model [9]. Previously, we identified that X01, 0315, and CSC2 cells had high invasive features, whereas 83NS and 528NS revealed a low invasiveness potential [21]. Therefore, by using highly invasive CSC2 and 0315 GSCs, the effects of PFD on cell migration and invasion were examined by Transwell assay. Migration of PFD-treated 0315 and CSC2 cells was significantly delayed by 50% and 45%, respectively, as compared to control cells (Figure 3A, *p* ≤ 0.001). Then, we examined GSC invasion by a Matrigel-coated microfluidic device. The control 0315 and CSC2 GSCs collectively invaded the Matrigel barrier, as opposed to single cell invasion. In addition, the cell morphology showed a spindle shape with mesenchymal-like features. However, PFD treatment reduced the invasion distance by 45% and 46% in 0315 and CSC2 cells, respectively (Figure 3B, ******
*p* ≤ 0.01, *******
*p* ≤ 0.001). For 3D invasion assays, we implanted 0315 GSC spheres in a Matrigel matrix. The implanted 0315 GSC sphere migrated radially into the Matrigel matrix, exhibiting an in vivo tumor behavior. After 72 h of treatment with 2 µM PFD, invasion from the tumor sphere exhibited a 48% decrease compared to the untreated 0315 GSCs (Figure 3C). To further confirm these observations, we examined various invasion-related molecules. We identified that the PFD treatment reduced expression of Integrin α6 and uPAR in CSC2 and 0315 GSCs at mRNA and protein levels (Figure 3D,E). In addition, the immunofluorescence analysis revealed the reduction of uPAR expression by 3 µM PFD treatment in CSC2 sphere-forming cells (Figure 3F).

### 2.4. PFD Suppresses Epithelial-Mesenchymal Transition Induced by GLI1 in GSCs

Epithelial-mesenchymal transition (EMT) is strongly associated with increased cell motility as well as resistance to genotoxic agents, which are typical characteristics of GSCs [22]. Previously, we found that PFD inhibits GSC proliferation and invasion, suggesting that PFD may suppress EMT factors. To elucidate the effect of PFD on EMT factors, we examined mRNA expression of GSCs 0315, CSC2, and X01 after 2 μM PFD treatment. Interestingly, PFD decreased the mRNA levels of vimentin, Zeb1, N-cadherin, Snail, and Slug, depending on the GSC type (Figure 4A). In addition, the western blot analysis identified decreased expression of EMT markers following PFD treatment (Figure 4B). Furthermore, we established GLI1 as a regulator of EMT factors. PFD inhibited GLI1 expression in a dose-dependent manner (Figure 4B). To identify GLI1 as a regulator of EMT factors in GSCs, it was knocked down in CSC2 cells by GLI1 siRNA. We identified that knockdown of GLI1 caused reduction in the mRNA and protein levels of vimentin, Zeb1, N-cadherin, Snail, and Slug in CSC2 cells (Figure 4C,D). These findings also confirmed the results of immunofluorescence staining, which showed that PFD decreased GLI1 and vimentin expression (Figure 4E).

### 2.5. PFD in Combination with TMZ Significantly Reduces GBM Growth In Vivo

The antitumor effect of PFD was further evaluated in combination with TMZ in a mouse orthotopic xenograft model. We implanted 1 × 10^4^ CSC2 cells into the brain parenchyma of nude mice. Two days later, mice were randomly divided into four groups (*n* = 5 mice/group): control, PFD, TMZ, and PFD combined with TMZ. PFD was administered daily, 0.8 mg/kg/week, by oral gavage for 28 days, TMZ was orally administered a dosage of 33 mg/kg/3 days per week, for 28 days; the PFD and TMZ combined treatment group was also administered the same dose, whereas mice in the control group received normal saline only during the experimental period (until day 60). Weight of mice was measured every 5 days (Figure 5A). We identified that all the mice survived for 60 days in TMZ-treated and the combined treatment group, compared with only 36 days in the control group, and 52 days in the PFD group (Figure 5B). To investigate whether PFD combined with TMZ inhibits CSC2 growth in vivo, we measured tumor volumes in xenograft models. The PFD treatment group showed decreased invasion and 30% reduced tumor volume compared to control. In addition, the combined treatment group inhibited all growth, and tumors disappeared in mice altogether, although the TMZ treatment group revealed 80% decrease in size of tumors (Figure 5C). To determine the robust effect of combined therapy, we performed immunohistochemical staining for the expression of GLI1, uPAR, SOX2, Vimentin, and Ki-67 in each group (Figure 5D). Interestingly, the residual tumors in the TMZ treatment group revealed the expression of GLI1, uPAR, SOX2, Vimentin, in contrast to the PFD treatment group. But, Ki-67 expression is similar in both group.

## 3. Discussion

In the present study, we identified the antitumor activity of PFD against GSCs, which are known to be one of the important determinants of resistance against chemo- and radiation therapies and cancer recurrence in human glioblastomas. The PFD treatment suppressed the proliferation, stemness, and invasion in GSCs, along with the inhibition of SOX2, uPAR, and GLI1 related EMT factors, which are known to be enriched in GSCs, and are critical for tumor maintenance [23,24,25]. We confirmed the efficacy of PFD in an orthotopic animal model. Interestingly, PFD suppressed the growth of glioblastoma and tumor invasion across the contralateral parenchyme, suggesting that PFD can easily penetrate the blood–brain barrier, and has an antitumor effect. In addition, a combination treatment of PFD with TMZ did not show tumor growth in orthotopic mouse brain. Hence, the results indicate that PFD plays an essential role in controlling the behavior of human glioblastoma sphere-forming cells, and that it may be advantageous to use in combination with TMZ. 

To repurpose drug for glioblastoma, several factors should be considered, including blood-brain barrier penetration, drug toxicity, and resistance to conventional chemotherapeutic agents. From this perspective, antipsychotic or antidepressant drugs can serve as an excellent candidate for glioblastoma control. These drugs are known to be commercially available for various psychiatric diseases and easily penetrate the blood-brain barrier. In addition, antipsychotic drugs, such as clozapine, chlorpromazine, and haloperidol and antidepressants, like imipramine, can regulate the Hedgehog pathway [26]. The Hedgehog signaling pathway plays a critical role in glioblastoma tumor progression and pathogenesis [27]. As an effector molecule of the Hedgehog pathway, GLI1 is known to be increased in glioblastomas, and the expression level of GLI1 is strongly associated with patient survival according to the CGGA mRNA array and RNA sequencing data [28]. GLI1 enhances cancer stem cell properties by upregulating stem cell markers like OCT4, Nanog, and SOX2. In addition, the resistance of glioblastoma to current therapies is associated with GLI1 because hedgehog-GLI signaling regulates the expression of stemness genes [24]. In this study, the proliferation properties of GSCs did not significantly decrease in a therapeutic range of TMZ, although MGMT protein was not expressed; PFD suppressed cell proliferation, stemness, and tumor invasion in various GSCs. PFD treatment reduces GSC proliferation; expression of stemness markers, including SOX2, OCT4, and NESTIN; and invasiveness markers, including Integrin α6, and uPAR [19,20,29]. The urokinase receptor is known to be a membrane-bound uPA receptor that plays a role in survival, angiogenesis, migration, and metastasis of GBM. In addition, PFD reduces the expression of EMT-related genes, including vimentin, Zeb1, N-cadherin, and Snail, caused by GLI1 inhibition [28,30,31]. Furthermore, PFD has an effect on immune modulation in glioblastoma. PFD treatment does not only suppress regulatory T cells but increases M1 macrophages. CCL4 and IFNγ, inflammatory markers associated with tumor progression, were also reduced with PFD treatment [32]. 

Some antipsychotic drugs surpass the therapeutic ranges of dosage in the case of human cancer treatment, although they were reported to have antitumor effects and no adverse effect in animal studies [16,19]. However, antipsychotic drugs may accumulate in other tissues and cause involuntary movement disorders or non-neuronal adverse effects, including blood sugar change, gastrointestinal discomfort, and hormonal imbalance [33]. Therefore, while repurposing drugs, care should be taken not regarding the tumoricidal effect of the drug but also with respect to proper dosage in cancer management. PFD, a diphenylbutylpiperidine derivative, is a long-acting antipsychotic drug, and the recommended dose in humans is 20 to 60 mg/week for better compliance [34]. PFD is sometimes well tolerated up to 160 to 250 mg/week in severely ill psychiatric patients [19]; however, others reported that PFD could induce extrapyramidal symptoms at a dosage of 40 mg/week [35]. Therefore, a new strategy needs to be determined while considering PFD regimen in cancer. In this study, we administrated 0.8 mg/week of PFD in orthotopic mouse glioblastoma model, which is the dose equivalent to 4 mg/week in a 60 kg adult. For estimating the safe starting dose in clinical trials for therapeutics in adult healthy volunteers, the following website was used: https://www.fda.gov/regulatory-information/search-fda-guidance-documents/estimating-maximum-safe-starting-dose-initial-clinical-trials-therapeutics-adult-healthy-volunteers. Therefore, the dosage of PFD in our experiment is within an affordable range for clinical use. 

In our animal study, we found that there was a significant reduction of tumor size and invasion adjacent to the normal brain in PFD treatment, although TMZ is a more effective chemotherapeutic drug to control glioblastoma. Immunostaining revealed that there was significantly decreased expression of GSC-related factors, including GLI1, SOX2, and vimentin in the background of PFD treatment, but residual tumor in TMZ treatment group still exhibited the expression of those proteins. This suggests that PFD demonstrates an effective antitumor activity through the inhibition of GSC-related factors, which did not respond to the conventional chemotherapeutic drug, TMZ [36]. For this reason, PFD combined with TMZ showed maximal antitumor effect and increased survival, suggesting that glioblastoma can be effectively controlled using this combination therapy. 

## 4. Materials and Methods 

### 4.1. Cell Culture and Drugs Treatment

Five GSCs CSC2, X01, 0315, 528NS, 83NS were used in this study. Cells were cultured in a proliferation medium composed of DMEM/F-12, B-27 supplement (Gibco, Waltham, USA), 10 ng/mL recombinant human bFGF (Pepro Tech, Rocky Hill, USA), 20 ng/mL recombinant human EGF (R&D systems, Minneapolis, MN, USA), 20 UI/mL penicillin, and 20 μg/mL streptomycin (complete medium). Two GBM cell lines (U87MG, T98G) were cultured in DMEM supplemented with 10% fetal bovine serum. 

PFD and TMZ (Sigma, St. Louis, MI, USA) were dissolved in DMSO to prepare a stock concentration of 20 and 100 mM, and then, diluted to the required concentrations with complete cell culture medium. The final concentration of DMSO was <0.1%, which had no effect on cell viability measured by MTT assay. Dose-response studies were carried out to determine the suitable doses for further experiments. Cell culture treatments were assessed following a schedule of administration: drug treatment was performed using 0.5–20 μM PFD for 24, 48, and 72 h and 50–600 μM TMZ for 120 h.

### 4.2. Immunoblotting and Antibodies

Whole-cell lysates were prepared from glioma cell and GSCs. Equivalent amounts of lysates were separated by 10% SDS polyacrylamide gel electrophoresis and transferred to polyvinylidene difluoride membranes. After blocking with 3% skim milk in Tris-buffered saline with Tween 20 for 0.5 h at room temperature (RT), membranes were incubated with diluted anti-NESTIN, OCT4, E-cadherin, N-cadherin, Vimentin, SOX2, Integrin α6, Slug, Zeb1, Snail, GLI1, uPAR (Cell signaling), and anti-β-ACTIN (Santa Cruz) primary antibodies. Horseradish peroxidase-conjugated anti-mouse or anti-rabbit secondary antibodies were used, and bound antibodies were detected using the ECL system.

### 4.3. Reverse Transcription and Quantitative PCR (RT-qPCR)

Total RNA was extracted using Trizol reagent (Invitrogen), according to the manufacturer’s instructions. Total cDNA was reverse transcribed from 1 μg of total RNA. To quantify gene expression, two-step qRT-PCR was performed using iQ SYBR Green supermix in Bio-Rad CFX96 real-time PCR detection system (Bio-Rad, Richmond, CA, USA). Expression levels were normalized to GAPDH. All qRT-PCR data were analyzed using the modified ΔΔCT method.

### 4.4. Primer

The following primers were used in Table 1:

### 4.5. Migration and Invasion Assay

These assays were performed using the BD BioCoat Tumor Invasion System. FluoroBlok 24-Multiwell Insert Plate with an 8 μm pore size PET membrane was used for migration assay, and the same insert plate uniformly coated with BD Matrigel Matrix (BD Biosciences) was used for the invasion assay. Sorted 1 × 10^6^ CSC2 cell subpopulations in 500 μL serum-free medium were seeded on the apical chamber, and then, 750 μL of chemoattractant (10% FBS in DMEM) was added to each of the basal chambers. After 48 h incubation, cells migrated to the lower chamber were stained with 0.5% crystal violet in PBS for 1 h at 37 °C and 5% CO_2_. Then, cells in five representative microscopic fields were counted and photographed.

### 4.6. Invasion Assay in the Microfluidic Device

The microfluidic device was fabricated by a traditional soft lithographic method as per Shin et al. [37]. The 10 μL of mixed Matrigel and culture medium (9:1) was gently injected into the middle channel of the device and incubated for 30 min in a 37 °C CO_2_ incubator for thermal crosslinking. Harvested cells were prepared at a density of 5 × 10^5^ cells in 50 μL of culture medium and seeded onto one side channel. The other channel was filled with 50 μL of culture medium, and the device was tilted and incubated for cell settlement and attachment to the wall of Matrigel construction for 2 h. Unattached cells were washed, and the channel was filled with fresh culture medium. For another case of invasion from GSC spheroid, the middle channel was filled with a mixture of Matrigel and culture medium (9:1) containing pre-cultured GSC spheroids in C-Well (300 µm, INCYTO, Cheonan, Korea) for 3 days. The device was incubated in 37 °C CO_2_ incubator for 30 min, and both side channels were filled with 120 µL of fresh culture medium or 2 μM PFD culture medium. Medium change was performed every 12 h with fresh culture medium.

### 4.7. Sphere Formation Assay

Tumorsphere formation assays were performed in a manner similar to our prior report [21]. Tumor spheres of 0315, 528NS, CSC2, and X01 cells were maintained in DMEM/F-12, B-27 supplement, 10 ng/mL recombinant human bFGF, and 20 ng/mL recombinant human EGF, and 20 UI/mL penicillin and 20 μg/mL streptomycin. Sphere-forming cells were plated on 6-well culture plate. Spheres were collected from culture plate along with media. Sphere pellet was formed by centrifugation, re-suspended in Accutase (Gibco), and placed in the incubator for 3 min. Pellet was pipetted to break the tumor spheres. Cells were counted, and approximately 1 × 10^5^ cells per well were plated on 6-well plate. After incubation, cells from tumor sphere were treated with 0, 1, 2, and 3 μM PFD for another 48 h. Images were obtained using phase-contrast microscopy. For in vitro limiting dilution assay, GSCs with decreasing numbers of cells (100, 50, 25, 12, 6, and 3) per well plated in 96-well plates containing DMEM/F-12 with B27, EGF (10 ng/mL), and bFGF (5 ng/mL) were used. Extreme limiting dilution analysis was performed using software available at http://bioinf.wehi.edu.au/software/elda/ [38].

### 4.8. GLI1 Silencing

CSC2 cells were transfected with GLI1 siRNA (Santa Cruz Biotechnology, Inc., Dallas, TX, USA) using RNAimax (Thermofisher, Waltham, MA, USA) transfection reagent, as per the manufacturer’s protocol. Cells were transfected with 100 nM GLI1 siRNA, and again after 12 h post-transfection. The cells were processed for western blotting and qRT-PCR analysis, as described before.

### 4.9. Immunocytochemistry

Tumor spheres were transferred from a T25 flask to 5 mL tube using a 1000 μL blunt end tip. Harvested spheres were centrifuged at 500× *g* for 3 min, and the supernatant was discarded. After washing with PBS, the spheres were fixed with 200 μL of 4% paraformaldehyde in PBS for 20 min at RT, washed with 800 μL of PBS, and left for 3 min. They were then centrifuged at 500× *g* for 3 min, which was repeated three times. After aspiration of the PBS, the fixed spheres were incubated with 200 μL of 0.025% (*w*/*v*) Triton X-100 at RT for 5 min, washed with 800 μL of PBS, and left for 3 min. They were then again centrifuged at 500× *g* for 3 min, which was repeated three times. After aspiration of the PBS, the spheres were blocked by incubation in 200 μL of PBS containing 3% FBS for 30 min, washed with 800 μL of PBS, and left for 3 min. They were then centrifuged at 500× *g* for 3 min, which was repeated three times. Spheres were subsequently processed for immunofluorescence staining using various antibodies. The spheres were incubated with the primary antibody in 200 μL of PBS for 1h at RT, or overnight at 4 °C. Spheres were washed with 800 μL of PBS, and then, left for 3 min. Again, they were centrifuged at 500× *g* for 3 min, which was repeated three times. Monoclonal antibodies against GLI1 (1:100 dilution), SOX2 (1:200 dilution), uPAR (1:200 dilution), and vimentin (1:200 dilution) were prepared. After aspiration of the PBS, the spheres were then incubated with the secondary antibody in 200 μL of PBS for 1 h at RT. Spheres were washed with 800 μL of PBS, and then, left for 3 min. Once again, they were then centrifuged at 500× *g* for 3 min, which was repeated three times. The secondary antibodies used in this protocol were cy3 anti-mouse, anti-rabbit, anti-goat IgG (1:200 dilution) with Hoechst 33342 (1:500 dilution), and the spheres were observed using confocal microscopy.

### 4.10. In Vivo Tumor Models

All procedures were conducted in accordance with the guidelines and protocol approved by the Institutional Animal Care and Use Committee (IACUC) of the Korea University College of Medicine. Five- to six-week-old female Balb/c nude mice were purchased from Orient Bio. CSC2 cell line (1 × 10^4^ cells) were suspended in 5 μL PBS and injected through the intracranial route using a stereotaxic device (David Kopf Instruments). Two days after tumor implantation, 20 mice were randomly assigned to four groups with 5 mice in each group to be treated with different drugs, as follows:Group 1 (vehicle): daily oral administration of normal saline;Group 2: daily oral administration of PFD, 0.8 mg/kg/week from day 3 to 30;Group 3: oral administration of TMZ, 33 mg/kg/5 days/week, 3 cycles from day 3 to 24;Group 4: TMZ and PFD combination for up to day 30.

The experiment was terminated at day 60, one month after the last injection, and animals were sacrificed by CO_2_ inhalation. Then, the brain was removed, fixed in 4% paraformaldehyde, and paraffin-embedded. The tissue was serially sectioned (4 μm) and stained with Mayer’s hematoxylin and eosin (Sigma–Aldrich, St. Louis, USA). Tumor volumes, greatest longitudinal diameter (length), and greatest transverse diameter (width) were measured using an external caliper and Adobe Illustrator software. The tumor volumes were calculated by the following formula: volume = 1/2 (length × width^2^) [39].

This study was approved by the IACUC of Korea University college of medicine (approval number KOREA-2016-0100 and KOREA-2019-0006).

### 4.11. Immunohistochemistry

Formalin-fixed and paraffin-embedded mouse tissue samples were sectioned, deparaffinized in xylene, hydrated in graded ethyl alcohols, and rinsed with PBS. For epitope antigen retrieval, the 5 μM-thick sections were soaked in pH 6.0 citrate buffer and heated in a microwave oven for 8 min. The sections were cooled for 15 min on ice, and endogenous peroxidase activity was blocked using hydrogen peroxide, methanol, and 10% fetal bovine serum for 30 min. The sections were incubated with primary antibodies GLI1, uPAR, SOX2, vimentin, and Ki-67 (1:100), and then, with horseradish peroxidase (HRP)-conjugated secondary antibodies. Immunoreactivity was detected with diaminobenzidine chromogen, and cell nuclei were counterstained with hematoxylin and positive staining area was evaluated with imageJ [40].

### 4.12. Statistical Analyses

The results are expressed as mean ± standard deviation (SD). The significance of the results was determined by the Student’s *t*-test (2-tailed), Mann-Whitney U test, and the Log-rank test by using SPSS 24 program (SPSS, INC., Chicage, IL, USA). A *p* value < 0.05 was considered significant.

## 5. Conclusions

The present work provides convincing data that PFD suppresses stemness, invasiveness, and EMT potential in human GSCs and that the antitumor effects of PFD are mediated at least in part by GLI1 inhibition. As several clinical trials using GLI1 inhibitors, like GANT61, have a limitation with respect to crossing the blood–brain barrier and causing unwanted adverse effects [41], therefore, PFD may be a potential candidate drug for glioblastoma treatment, especially in combination with TMZ. 

## Figures and Tables

**Figure 1 cancers-11-01310-f001:**
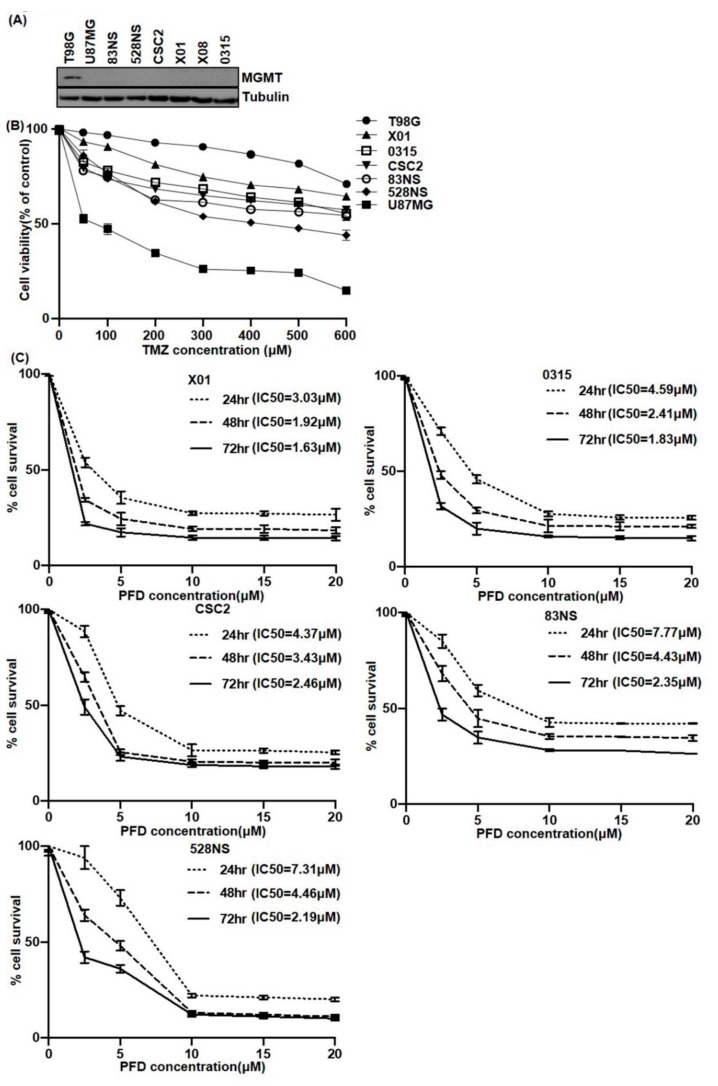
GSCs and glioma cell proliferation upon TMZ and PFD treatment. (**A**) MGMT protein expression in T98, U87MG glioma cells, and 6 GSCs by western blot. Tubulin was used as loading control. (**B**) T98, U87MG glioma cells, and X01, CSC2, 528NS, 83NS, and 0315 GSCs were treated with different concentrations of TMZ for 5 days. Cell proliferation was measured by MTT assay. The experiments were repeated three times. (**C**) 0315, CSC2, 528NS, X01, and 83NS GSCs were treated with different concentrations of PFD for 24, 48, and 72 h. Cell proliferation was measured by MTT assay to estimate the IC50 values. The experiments were repeated three times.

**Figure 2 cancers-11-01310-f002:**
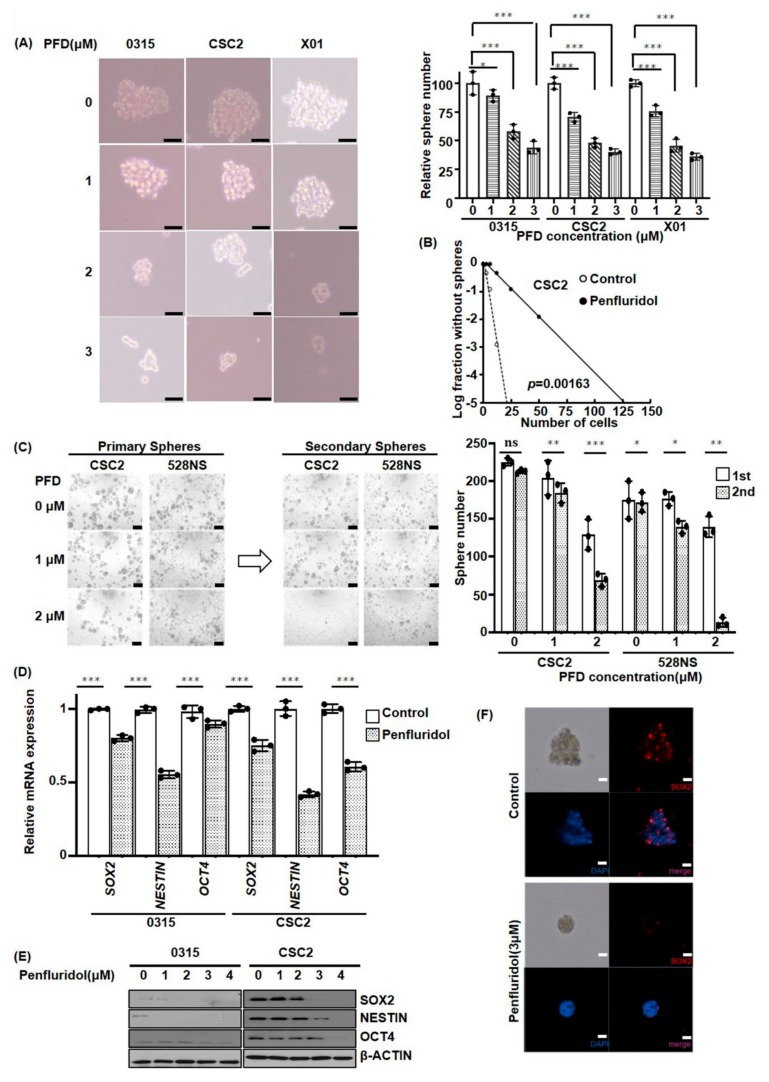
PFD suppresses the stemness of GSCs. (**A**) Sphere formation was tested in 0315, CSC2, and X01 GSCs. Following 1, 2, and 3 µM PFD treatment, the number of sphere formations was counted by phase contrast microscopy at 48 h (* *p* < 0.05, *** *p* ≤ 0.001, when compared with no treatment, Scale bar: 100 µm). (**B**) Limiting dilution assay was performed in CSC2 cells following 2 µM PFD treatment or its control. (*p* = 0.00163) (**C**) The 528 and CSC2 GSCs were treated with 1 and 2 μM PFD, respectively. Twenty-four hours later, the number of sphere formations was counted by phase contrast microscopy (Scale bar: 100 µm). The same procedure was then repeated, and secondary sphere formation was counted (ns; no significance, * *p* < 0.05, ** *p* ≤ 0.01, *** *p* ≤ 0.001, when compared with no treatment). (**D**, **E**) RT-qPCR and western blotting analyses for stemness factors, including SOX2, OCT4, and NESTIN genes normalized to GAPDH or β actin, in response to PFD treatment. Data are presented as the mean ± SD of triplicate qPCR assays (*** *p* ≤ 0.001, when compared with no treatment). (**F**) Immunocytochemistry for the stemness factor, SOX2 in CSC2 cells treated with 3 μM PFD for 48 h (blue, nucleus; red, SOX2, Scale bar: 100 µm).

**Figure 3 cancers-11-01310-f003:**
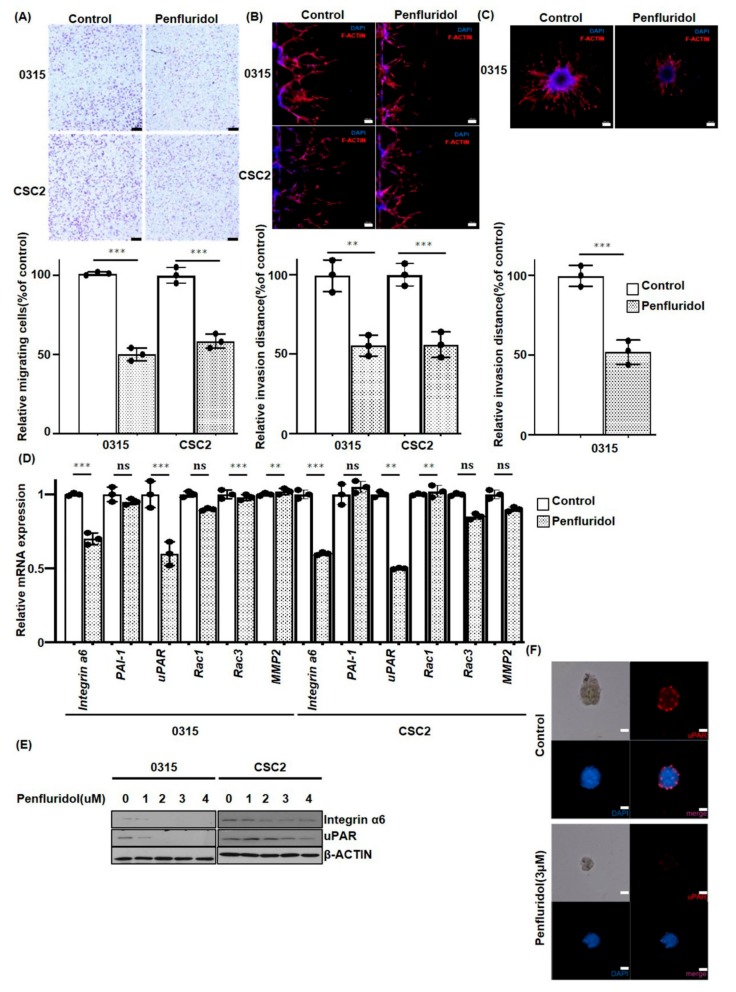
PFD suppresses GSCs migration and invasive potential. (**A**) 0315 and CSC2, highly invasive GSCs, were seeded in the Transwell and treated with 2 µM PFD. At 24 h, traversed cells were stained and counted (*** *p* ≤ 0.001). No treatment was used as control. The experiment was repeated three times, and values are plotted as mean ± SD (Scale bar: 100 µm). (**B**) 0315 and CSC2 GSCs were attached to the microfluidic device and treated with 2 μM PFD. Cells invading the other channel were stained with F-actin and counted at 48 h. Invading cells in PFD treatment group were compared with control (** *p* ≤ 0.01, *** *p* < 0.001). The experiment was repeated three times, and values were plotted as mean ± SD (Scale bar: 100 µm). (**C**) For 3D invasion assay from a GSC spheroid in the microfluidic device, 0315 GSCs were treated with 2 µM PFD. At 72 h, invaded spheroids were stained with F-actin, and calculated in comparison to the control (*** *p* < 0.001). Experiment was repeated three times, and values are plotted as mean ± SD (Scale bar: 100 µm). (**D**) RT-qPCR for Integrin α6, PAI-1, uPAR, Rac1, Rac3, and MMP2 at 24 h after 2 μM PFD treatment, and (**E**) Western blotting analyses of Integrin α6 and uPAR in CSC2 and 0315 cells. GAPDH or β-actin were used as loading control. Data are presented as the mean ± SD of triplicate qPCR assays (ns, no significance; ** *p* < 0.01, *** *p* < 0.001, when compared with no treatment). (**F**) Immunocytochemistry for uPAR in CSC2 cells with 3 μM PFD treatment at 48 h (blue, nucleus; red, uPAR, Scale bar: 100 µm).

**Figure 4 cancers-11-01310-f004:**
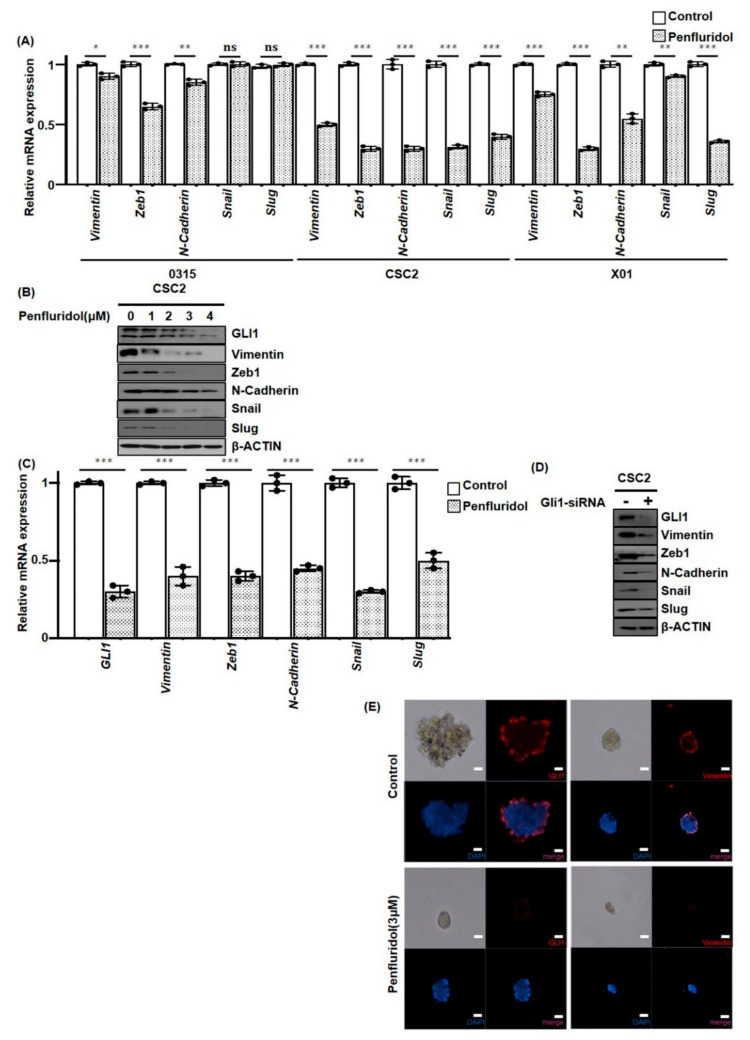
GLI1 inhibition by PFD suppresses EMT factors in GSCs. (**A**) RT-qPCR for vimentin, ZEB1, N-cadherin, Snail, and Slug at 48 h after 2 μM PFD treatment, and (**B**) Western blotting analyses for GLI1, vimentin, ZEB1, N-cadherin, Snail, and Slug at different concentrations of PFD. Loading control used was GAPDH or β-actin. Data are presented as the mean ± SD of triplicate qPCR assays (ns, no significance; * *p* < 0.1, ** *p* < 0.01, *** *p* < 0.001, when compared with no treatment). (**C**,**D**) CSC2 cells were transduced with GLI1 siRNA, and examined by RT-qPCR and western blot analyses. Data are presented as the mean ± SD of triplicate qPCR assays (*** *p* < 0.001). (**E**) Immunocytochemistry for GLI1 and vimentin in CSC2 cells with 3 μM PFD treatment at 48 h (blue, nucleus; red, vimentin and GLI1, Scale bar: 100 µm).

**Figure 5 cancers-11-01310-f005:**
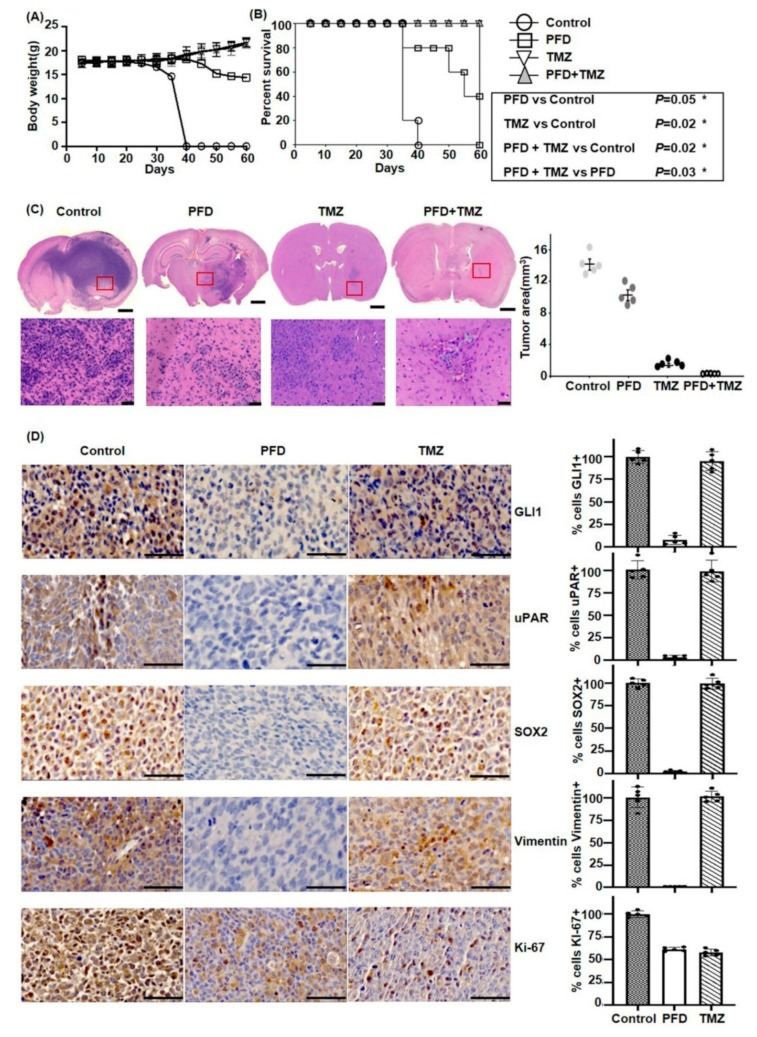
PFD exhibits antitumor effect and increases survival in vivo. (**A**) The body weight of mice after administration of different formulations of PFD and TMZ up to day 60 (*n* = 5). (**B**) Kaplan‒Meier survival curve of mouse implanted with 1 × 10^4^ CSC2 GSCs. Mice were treated with PBS control, 0.8 mg/week PFD, 33 mg/kg TMZ, and PFD + TMZ combination (*n* = 5, log-rank test results are shown in the inset). (**C**) Histopathology of mouse brains obtained after 40 days post-implantation. The tissues were stained with H&E to detect tumor size (Scale bar: 1000 µm) and invasiveness (Scale bar: 50 µm). Tumor volume in mice was measured by Image Tool 3.00 (*** *p* ≤ 0.001, One-way ANOVA, Student-Newman–Keuls post hoc test). (**D**) Tumor tissues were sectioned and immunostained for detecting the expression of GLI1, SOX2, uPAR, Vimentin and Ki-67. Representative images are shown and data are presented as mean ± SD (Scale bar: 50 μm).

**Table 1 cancers-11-01310-t001:** Primer list.

SOX2	F	AGAAGAGGAGAGAGAAAGAAAGGGAGAGA
	R	GAGAGAGGCAAACTGGAATCAGGATCAAA
NESTIN	F	CAGCGTTGGAACAGAGGTTGG
	R	TGGCACAGGTGTCTCAAGGGTAG
OCT4	F	AACCTGGAGTTTGTGCCAGGGTTT
	R	TGAACTTCACCTTCCCTCCAACCA
Integrin-a6	F	GTGGCTATTCTCGCTGGGAT
	R	ACCTAGAGCGTTTAAAGAATCCAC
PAI-1	F	ACAAGTTCAACTATACTGAGTTCACCACGCCC
	R	TGAAACTGTCTGAACATGTCGGTCATTCCC
uPAR	F	CCTCTGCAGGACCACGAT
	R	TGGTCTTCTCTGAGTGGGTAC
Rac1	F	CTGAAGGAGAAGAAGCTGAC
	R	TCGTCAAACACTGTCTTGAG
Rac3	F	GACGACAAGGACACCATTGA
	R	CCTCGTCAAACACTGTCTTC
MMP2	F	GATAACCTGGATGCCGTCGT
	R	CGAAGGCAGTGGAGAGGAAG
GLI1	F	CAACTCGATGACCCCACCAC
	R	CAGACAGTCCTTCTGTCCCCA
Vimentin	F	CCAGGCAAAGCAGGAGTC
	R	CGAAGGTGACGAGCCATT
Zeb1	F	GTGGCGGTAGATGGTAAT
	R	CTGTTTGTAGCGACTGGA
N-Cadherin	F	TGCAAGACTGGATTTCCTGAAGA
	R	AGCTTCTCACGGCATACACC
Snail	F	CGGAAGCCTAACTACAGCGA
	R	GCCAGGACAGAGTCCCAGAT
Slug	F	AGATGCATATTCGGACCCAC
	R	CCTCATGTTTGTGCAGGAGA
HPRT1	F	AAAGGACCCCACGAAGTGTT
	R	AAGCAGATGGCCACAGAACTA

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
