# Peer review of "Repurposing Penfluridol in Combination with Temozolomide for the Treatment of Glioblastoma"

_cancers, 2019, doi:10.3390/cancers11091310_

Round 1

Reviewer 1 Report

The manuscript provides an interesting argument to repurpose Penfluidol for use in GBM treatment. The following points should be addressed before it is ready for publication:

- What is the molecular target of penfluidol and its expression in GBM/CSCs? Please provide data from public datasets (e.g. TCGA, Rembrandt…) as well as in vitro confirmation.

- The sphere formation assays (Fig 2A-B) were not performed per the standard of the field. Please see https://doi.org/10.1016/j.jim.2009.06.008, https://doi.org/10.1016/j.cell.2011.06.006, and https://doi.org/10.1093/neuonc/noy012 for examples of a proper assay. The point of the assay is to measure self-renewal frequency, which is not clear if done as in the manuscript. Please perform the assays one more time, with proper citations.

- Regarding the expression of stem cell markers (Fig 2C-D), does penfluridol affect the expression of other well-accepted markers such as CD133 and CD44?

- For in vivo experiment, please provide growth curves (e.g. measured by luciferase bioluminescence) beside the endpoint tumor size. Furthermore, if the authors wanted to make a point about cancer stem cells, 10000 cells/animal were too many. Please see https://doi.org/10.1093/neuonc/noy012 for an example of such case.

- Please also examine tumor cell infiltration by histological staining in the in vivo sample, to match the in vitro data.

- Regarding figures:

+ Fig 1B: It is hard to distinguish between T98G and U87MG. Please consider changing colors/markers to make it more readable.

+ Fig 1C: Please make the medium markers and error bars more visible.

+ For bar graphs, please plot the actual data points. Please see https://www.graphpad.com/support/faq/graph-tip-how-can-i-make-a-barcolumn-graph-that-also-shows-the-individual-data-points/

+ Fig 2A, 2B, 2E, 3A-C, 3F, 4E, 5B: Please add scale bars and provide proper captions.

Author Response

Reviewer2

Comment 1: What is the molecular target of penfluidol and its expression in GBM/CSCs? Please provide data from public datasets (e.g. TCGA, Rembrandt…) as well as in vitro confirmation.

Response: We agree with this legitimate point. Identification of molecular targets of novel drugs is important to determine antitumor effect. Previous reports showed that penfluridol demonstrates the suppression of multiple target molecules including integrin alpha, GLI1 and myeloid derived suppressor cells in glioma cells(1-3). In addition, our study found that penfluridol also suppresses uPAR and EMT-related factors as well as stem cell factors. Among them, we think that a critical target molecule is a GLI1. GLI1 is known to regulate stem cell factors and our report shows that GLI1 regulates EMT-related factors. Therefore, we identified a genomic papers for GLI1 expression and clinical implication in glioblastoma which HEDGEHOG-GLI1 signaling regulates human glioma growth, cancer stem cell self-renewal and patient survival(4,5). Considering public genome data for GLI1, therefore, we described it in discussion section as like “GLI1 is over expressed in glioblastoma, and the expression level of GLI1 is strongly associated with patient survival in the CGGA mRNA array and RNA sequencing data”.

References)

Ranjan A, Gupta P, Srivastava SK (2016) Penfluridol: An Antipsychotic Agent Suppresses Metastatic Tumor Growth in Triple-Negative Breast Cancer by Inhibiting Integrin Signaling Axis. Cancer Res 76: 877-890. Ranjan A, Srivastava SK (2017) Penfluridol suppresses glioblastoma tumor growth by Akt-mediated inhibition of GLI1. Oncotarget 8: 32960-32976 Ranjan A, Wright S, Srivastava SK (2017) Immune consequences of penfluridol treatment associated with inhibition of glioblastoma tumor growth. Oncotarget. 18;8(29):47632-47641. Clement V, Sanchez P, de Tribolet N, Radovanovic I, Ruiz i Altaba A (2007) HEDGEHOG-GLI1 signaling regulates human glioma growth, cancer stem cell self-renewal, and tumorigenicity. Curr Biol 17: 165-172. Li J, Cai J, Zhao S, Yao K, Sun Y, Li Y, Chen L, Li R, Zhai X, Zhang J, Jiang C (2016) GANT61, a GLI inhibitor, sensitizes glioma cells to the temozolomide treatment. J Exp Clin Cancer Res 35: 184

Comment 2: The sphere formation assays (Fig 2A-B) were not performed per the standard of the field. Please see https://doi.org/10.1016/j.jim.2009.06.008, https://doi.org/10.1016/j.cell.2011.06.006, and https://doi.org/10.1093/neuonc/noy012 for examples of a proper assay. The point of the assay is to measure self-renewal frequency, which is not clear if done as in the manuscript. Please perform the assays one more time, with proper citations.

Response: We agree with this comment. Responding to this important point, we performed an additional experiment for sphere formation of GSCs by extreme limiting dilution analysis(ELDA), The new data is described in Methods and Results sections, respectively, with citations. However, due to time limitation given by Cancers editorial board (i.e. resubmission within 10 days after decision), we could perform experiment only with CSC2(Figure 2B). We hope this resolves the Comment 2 of Reviewer. Please see the attachment.

Comment 3: Regarding the expression of stem cell markers (Fig 2C-D), does penfluridol affect the expression of other well-accepted markers such as CD133 and CD44?

Response: Thank you for your critical point. As you know, there are several stem cell markers, including Sox2, Nestin, OCT4, CD44 and CD 133. It is known that

Hedgehog-GLI signaling regulates the self-renewal of CD 133+ glioma cancer stem cells whether it is direct effect on CD 133+ or not(1). However, in this study, we examined Sox2, Nestin and OCT4 expression, but did not identify CD44 and CD133 markers following penfluridol treatment

Reference)

Clement V, Sanchez P, de Tribolet N, Radovanovic I, Ruiz i Altaba A (2007) HEDGEHOG-GLI1 signaling regulates human glioma growth, cancer stem cell self-renewal, and tumorigenicity. Curr Biol 17: 165-172.

Comment 4: For in vivo experiment, please provide growth curves (e.g. measured by luciferase bioluminescence) beside the endpoint tumor size. Furthermore, if the authors wanted to make a point about cancer stem cells, 10000 cells/animal were too many. Please see https://doi.org/10.1093/neuonc/noy012 for an example of such case. Please see the attachment.

Response: We apologize for having failed to provide a growth curve which was measured by luciferase bioluminescence. We tried to produce luciferase-expressing stable cell lines, but failed. Therefore, we can just show in vivo tumor growth curve by measuring tumor and mice weight. We added the data for mouse weight in Figure 5A. Considering tumor cell amount for orthotopic mouse model, we injected 1x103 or 1x104 GSC cells to identify the tumorigenicity according to our previous publication (1). In this study, we wanted to focus on the antitumor effect of penfluridol and injected 1x104 CSC2 GSCs into the mouse brain. Please see the attachment.

Reference)

Yin J, Park G, Kim TH, et al. Pigment Epithelium-Derived Factor (PEDF) Expression Induced by EGFRvIII Promotes Self-renewal and Tumor Progression of Glioma Stem Cells. PLoS Biol. 2015 May 20;13(5):e1002152.

Comment 5: Please also examine tumor cell infiltration by histological staining in the in vivo sample, to match the in vitro data.

Response: We apologize for the ambiguity. Following your wonderful comment, we added the magnification figures for tumor cell infiltration in H&E examination. Please see the attachment.

Comment 6: Regarding figures:

+ Fig 1B: It is hard to distinguish between T98G and U87MG. Please consider changing colors/markers to make it more readable.

+ Fig 1C: Please make the medium markers and error bars more visible.

+ For bar graphs, please plot the actual data points. Please see https://www.graphpad.com/support/faq/graph-tip-how-can-i-make-a-barcolumn-graph-that-also-shows-the-individual-data-points/

+ Fig 2A, 2B, 2E, 3A-C, 3F, 4E, 5B: Please add scale bars and provide proper captions.

Response: We apologize for the low-quality images. We modified figures following your instruction and added scale bars. Please see the attachment.

Reviewer 2 Report

The manuscript entitled “Repurposing Penfluridol in Combination with Temozolomide for the Treatment of Glioblastoma” reports antitumor effects of Penfluridol on Glioma Sphere forming Cells (GSCs), thereby the authors propose the potential of repurposing Penfluridol as a combination treatment along with Temozolomide for glioblastoma.  They have evaluated the effects of Penfluridol on GSCs using in vitro assays and in vivo experiments. Treatment of Penfluridol on GSCs reduced the expression of stemness related genes, cell viability, and migration/invasion properties of these cells. They also demonstrated Penfluridol mediated inhibition of EMT associated factors. Finally, they demonstrated the in vivo anti-tumor effects of Penfluridol in combination with Temozolomide. The study is interesting, and the manuscript is presented systematically. The reviewer feels the manuscript is appropriate for publication in the journal

Few minor comments that the authors may address

Page 1. Line 2, typo in the title. It should be Penfluridol. Fig. 1C. In vitro cell proliferation assays were carried out to evaluate the effect of Penfluridol on the cell viability of GSCs. Did the authors investigate the combination treatment (Penfluridol+ Temozolomide) on these cell lines and its effect on cell proliferation? Fig. 5A. Include the survival analysis results for Penfluridol + TMZ vs TMZ in the table ad explain the findings in the results.  

Author Response

Comment 1: Page 1. Line 2, typo in the title. It should be Penfluridol.

Response: We apologize for the typo. We corrected the title to “Repurposing Penfluridol in Combination with Temozolomide for the Treatment of Glioblastoma”

Comment 2: Fig. 1C. In vitro cell proliferation assays were carried out to evaluate the effect of Penfluridol on the cell viability of GSCs. Did the authors investigate the combination treatment (Penfluridol+ Temozolomide) on these cell lines and its effect on cell proliferation?

Response: We thank Reviewer 1 for suggesting this important data. We performed in vitro cell proliferation assay to determine the effect of combination treatment in glioma cells. The new data is in supplementary Figure 1. Please see the attachment.

Comment 3: Fig. 5A. Include the survival analysis results for Penfluridol + TMZ vs TMZ in the table ad explain the findings in the results. 

Response: We apologize for incomplete description of the data. We added the analysis results of Penfluridol + TMZ vs TMZ in the Results section.

Before) We identified that the overall survival in mice was significantly increased in the combined group and TMZ-treated group, compared with only 36 days in the control group, and 52 days in the penfluridol group (Figure 5A).

After) We identified that all of mice were survived for 60 days in TMZ-treated and the combined treatment group, compared with only 36 days in the control group, and 52 days in the penfluridol group (Figure 5B).

Reviewer 3 Report

In this work, Kim and collaborators investigated the anti-tumor effect of penfluridol on glioma cells. They show that it inhibits several tumor-associated phenotypes such as self-renewal, migration, invasion, EMT. Furthermore, they show that combination treatment of penfluridol with temozolomide (TMZ) significantly suppressed tumor growth, and prolonged survival in vivo. They propose that penfluridol can be used as an actionable repurposing drug for glioblastoma. The manuscript requires major revision before can be accepted in Cancers.

MAJOR LIMITATIONS

The manuscript should be checked carefully to correct errors. There are many and very important (penfluidol in title, etc). The manuscript in its current format is below Cancers journal standards. The manuscript requires English Editing service. The manuscript in its current format is below Cancers journal standards. Experiments overexpressing Gli in Penfluridol treated cells should be done to confirm that this is a main mediator of Penfluridol. Moreover, ChIP assays should be done to test whether the effect is direct or not. Quantification of results of immunohistochemistry (Fig 5D) should be added. Moreover, immunostaining of Ki67 should be presented to reinforce at molecular level the antitumor activity of penfluridol, TMZ and combined treatment. Images of penfluridol treated tumors seem odd. Were all the tumors stained in parallel ? Weight of mice should be presented. Previous studies have been tested penfluridol in glioma cells with similar results (Ranjan & Srivastava 2017 Oncotarget https://www.ncbi.nlm.nih.gov/pubmed/28380428 Ranjan et al 2017 Oncotarget https://www.ncbi.nlm.nih.gov/pubmed/28512255). These works should be cited in the manuscript and authors should discuss the novel aspects of their results. This reviewer misses a general introduction and/or discussion regarding the possible impact of antipsychotic and glioblastoma (https://www.ncbi.nlm.nih.gov/pubmed/?term=antipsychotic+glioblastoma) and the advantages/disadvantages of penfluridol.

Author Response

Comment 1: The manuscript should be checked carefully to correct errors. There are many and very important (penfluidol in title, etc). The manuscript in its current format is below Cancersjournal standards. The manuscript requires English Editing service. The manuscript in its current format is below Cancers journal standards.

Response: We apologize for the quality of English. We had the manuscript read by professional English editors. Please see the revised manuscript.

Comment 2: Experiments overexpressing Gli in Penfluridol treated cells should be done to confirm that this is a main mediator of Penfluridol. Moreover, ChIP assays should be done to test whether the effect is direct or not.
Response: We appreciate your comment. In this study, we found that GLI1 inhibition suppressed expression of several EMT factors. GLI is a transcriptional factor and can regulate EMT factors as downstream targets (1-3). For this reason, it is important to determine whether GLI1 can transcriptionally regulate downstream targets or not. However, we did not have a chance to perform additional work for transcriptional regulation including overexpression GLI1 plasmid construction, promotor assay for each targets and Chip assays to determine target DNA by GLI1 protein binding. Currrently, we are planning to perform molecular works for GLI-induced regulation of EMT factors. Unfortunately, we were only given 10 days by the Cancers journal policy. Therefore, it is out of scope in this manuscript. Hopefully, we want to report this work in next paper.

Reference)

Li J, Cai J, Zhao S, Yao K, Sun Y, Li Y, Chen L, Li R, Zhai X, Zhang J, Jiang C (2016) GANT61, a GLI inhibitor, sensitizes glioma cells to the temozolomide treatment. J Exp Clin Cancer Res 35: 184 doi:10.1186/s13046-016-0463-3 Riaz SK, Ke Y, Wang F, Kayani MA, Malik MFA (2019) Influence of SHH/GLI1 axis on EMT mediated migration and invasion of breast cancer cells. Sci Rep 9: 6620 doi:10.1038/s41598-019-43093-x Katoh Y, Katoh M (2009) Hedgehog target genes: mechanisms of carcinogenesis induced by aberrant hedgehog signaling activation. Curr Mol Med 9: 873-886

Comment 3: Quantification of results of immunohistochemistry (Fig 5D) should be added. Moreover, immunostaining of Ki67 should be presented to reinforce at molecular level the antitumor activity of penfluridol, TMZ and combined treatment.

Response: We agree with your critical comment. Following our previous reports, we quantitated immunostain data(1). In addition, we immunostained Ki67 proliferation marker to determine antitumor effect of penfluridol by using orthotopic mouse tumor samples and added it in Result. Please see the attachment.

Reference)

Kim H, Park KJ, Ryu BK, et al. Forkhead box M1 (FOXM1) transcription factor is a key oncogenic driver of aggressive human meningioma progression. Neuropathol Appl Neurobiol. 2019 Jun 9.

Comment 4: Images of penfluridol treated tumors seem odd. Were all the tumors stained in parallel ? Weight of mice should be presented.

Response: We apologize for this confusion and absolutely agree with your opinion.

We restained mouse tumor samples and changed H&E stain images in parallel. Weight of mice was also added in Figure 5A. Please see the attachment.

Comment 5: Previous studies have been tested penfluridol in glioma cells with similar results (Ranjan & Srivastava 2017 Oncotarget https://www.ncbi.nlm.nih.gov/pubmed/28380428 Ranjan et al 2017 Oncotarget https://www.ncbi.nlm.nih.gov/pubmed/28512255). These works should be cited in the manuscript and authors should discuss the novel aspects of their results.

Response: Thank you for the critical comment. As you mentioned, Ranjan group reported papers for the effect of penfluridol in glioblastoma treatment. They found several suppressive effects for penfluridol including integrin alpha, GLI1 and myeloid derived suppressor cells in glioma cells. We already cited Ranjan’s paper, https://www.ncbi.nlm.nih.gov/pubmed/28380428. In this time, we added another paper, https://www.ncbi.nlm.nih.gov/pubmed/28512255, in Discussion as like below.

Furthermore, PFD(penfluridol) has an effect on immune modulation in glioblastoma. PFD treatment does not only suppress regulatory T cells, but increases M1 macrophages. CCL4 and IFNg, inflammatory markers associated with tumor progression, were also reduced with PFD treatment [34].

Ranjan A, Wright S, Srivastava SK (2017) Immune consequences of penfluridol treatment associated with inhibition of glioblastoma tumor growth. Oncotarget 8: 47632-47641 doi:10.18632/oncotarget.17425

 In addition, novel aspects of the our results have several critical points and are different from Ranjan’s works. First, we found that penfluridol has another novel targets including uPAR, EMT-related factors, and GSC associated factors. Second, it is known that standard chemotherapeutic treatment of glioblastoma is temozolomide at now. However, temozolomide has a limited effect on the patient survival and new therapeutic agent is urgently needed. Interestingly, we found that combination treatment (penfluridol and TMZ) increased antitumor effect in vivo experiment, compared to TMZ treatment. Penfluridol suppressed GSC associated factors in immunostain result and it may cause a robust effect in glioma control in combination with TMZ. Nobody reported this feature. Third, penfluridol has an antitumor effect to the suppression of GSC-associated factors in vivo animal model which is within therapeutic dosage in this study. It suggests that penfluridol can be an actionable drug in clinical trial. We described these points in discussion section. Please see the revised manuscript.

Comment 6: This reviewer misses a general introduction and/or discussion regarding the possible impact of antipsychotic and glioblastoma (https://www.ncbi.nlm.nih.gov/pubmed/?term=antipsychotic+glioblastoma) and the advantages/disadvantages of penfluridol.

Response: Glioblastoma is a lethal disease and a devastating median survival, range 12 to 14 months. However, current regimens have a critical limitation to improve survival. For this reason, actionable therapeutic modality is urgently needed. Our study focus on this and repurposing drug may be a good candidate. Among them, antipsychotics has an interesting points as a repurposing drug. Firstly, the antipsychotics can easily penetrate blood-brain barrier. In addition, several antipsychotics have been revealed antitumor effect for hedgehog signaling pathway and calcium channel modulation in various cancer types. Furthermore, some antipsychotics suppress GSC associated factors which are involved in the treatment resistance and tumor recurrence in glioblastoma(1-3). However, most antipsychotics surpass the therapeutic ranges although they revealed antitumor effect in vitro and in vivo animal tumor models(4). In our study, we identified antitumor effect of penfluridol in glioblastoma. Penfluridol suppresses GSC associated factors and effectively control in vivo glioblastoma growth within a therapeutic range. In addition, penfluridol increased tumor control in combination with TMZ. In conclusion, penfluridol may be a clinically actionable candidate drug for glioblastoma treatment, especially in combination with temozolomide. 

Reference)

Cheng HW, Liang YH, Kuo YL, Chuu CP, Lin CY, Lee MH, Wu ATH, Yeh CT, Chen EIT, Whang-Peng J, Su CL, Huang CYF (2015) Identification of thioridazine, an antipsychotic drug, as an antiglioblastoma and anticancer stem cell agent using public gene expression data. Cell Death Dis 6. Gilder AS, Natali L, Van Dyk DM, Zalfa C, Banki MA, Pizzo DP, Wang H, Klemke RL, Mantuano E, Gonias SL (2018) The Urokinase Receptor Induces a Mesenchymal Gene Expression Signature in Glioblastoma Cells and Promotes Tumor Cell Survival in Neurospheres. Sci Rep 8: 2982. Clement V, Sanchez P, de Tribolet N, Radovanovic I, Ruiz i Altaba A (2007) HEDGEHOG-GLI1 signaling regulates human glioma growth, cancer stem cell self-renewal, and tumorigenicity. Curr Biol 17: 165-172. Kang S, Hong J, Lee JM, Moon HE, Jeon B, Choi J, Yoon NA, Paek SH, Roh EJ, Lee CJ, Kang SS (2017) Trifluoperazine, a Well-Known Antipsychotic, Inhibits Glioblastoma Invasion by Binding to Calmodulin and Disinhibiting Calcium Release Channel IP3R. Mol Cancer Ther 16: 217-227.

Round 2

Reviewer 1 Report

The authors has revised their manuscript extensively and addressed my initial concerns. I only have one comment: Unless there is a constrain of the number of citations, please cite the method papers related to the sphere formation assay as I suggested in the original review. Citing papers for methodology provide incentive for their authors to continue developing new methods that we can use in our research. Afterward, the manuscript should be published without another round of review.

Author Response

Comment: Unless there is a constrain of the number of citations, please cite the method papers related to the sphere formation assay as I suggested in the original review. Citing papers for methodology provide incentive for their authors to continue developing new methods that we can use in our research. Afterward, the manuscript should be published without another round of review.

Response: Thank you for the comment. As you mentioned ELDA method paper, we added https://doi.org/10.1016/j.jim.2009.06.008 in material&method 

Reviewer 3 Report

Authors improved some of the indicated suggestions. 

Author Response

Comment: Authors improved some of the indicated suggestions.

Response: Thank you for the comment. As you mentioned at primary revision, we report Chip assay in next paper.
